# Tra-MoE: Scaling Trajectory Prediction Models for Adaptive Policy Conditioning

## Abstract

Scale is a primary factor that influences the performance and generalization of a robot learning system. In this paper, we aim to scale up the trajectory prediction model by using broad out-of-domain data to improve its robustness and generalization ability. Trajectory model is designed to predict any-point trajectories in the current frame given an instruction and can provide detailed control guidance for robotic policy learning. To handle the diverse out-of-domain data distribution, we propose a sparsely-gated MoE (**Top-1** gating strategy) architecture for trajectory model, coined as **Tra-MoE**. The sparse activation design enables good balance between parameter cooperation and specialization, effectively benefiting from large-scale out-of-domain data while maintaining constant FLOPs per token. In addition, we further introduce an adaptive policy conditioning technique by learning 2D mask representations for predicted trajectories, which is explicitly aligned with image observations to guide policy prediction more flexibly. We perform experiments on both simulation and real-world scenarios to verify the effectiveness of our Tra-MoE and adaptive policy conditioning technique. We jointly train the Tra-MoE model on all 130 tasks in the LIBERO benchmark and conduct a comprehensive empirical analysis, demonstrating that our Tra-MoE consistently exhibits superior performance compared to the dense baseline model, even when the latter is scaled to match Tra-MoE's parameter count.

## 1 Introduction

In computer vision (CV) and natural language processing (NLP), significant progress has been made through scaling up models from large-scale out-of-domain data. For example, many works use broad data to perform self-supervised pre-training, and then adapt to diverse downstream tasks through zero-shot (Brown, 2020; Radford et al., 2021; Achiam et al., 2023) or fine-tuning (Devlin, 2018; Chen et al., 2021; He et al., 2022; Tong et al., 2022). Additionally, some research efforts (Lu et al., 2022; Zhu et al., 2022b;a) focus on learning a generalist model capable of unifying various tasks. In contrast, in robot learning, due to data scarcity and homogeneity, some groundbreaking methods (Levine et al., 2016; Andrychowicz et al., 2020) resort to training only using in-domain data. Recently, some works (Seo et al., 2022; Bahl et al., 2023; Wang et al.; Bharadhwaj et al., 2024a) have shifted to learn from action-free human video data. A particularly scalable and promising paradigm (Wen et al., 2023; Xu et al., 2024; Bharadhwaj et al., 2024b) involves learning a trajectory prediction model from action-free video data, and then using small-scale action-labeled demonstrations to learn trajectory-guided policies, thereby achieving greater sample efficiency and better generalization.

Despite the potential benefits, it still remains underexplored how to effectively leverage broad out-of-domain video data for training trajectory models. In particular, the impact of incorporating data encompassing diverse skills, environments, objects, and even embodiments has not fully investigated for downstream tasks. For instance, ATM (Wen et al., 2023), a notable work in this field, mainly relies on in-domain data for training both trajectory models and policies. We argue that **scale** is the primary factor to build a powerful trajectory model for policy conditioning. Therefore, we propose to scale up the training of trajectory model under the setup of *jointly learning from large-scale out-of-domain and small-scale in-domain data*, as shown in Fig. 2(a). However, directly using large-scale out-of-domain video data for hybrid pre-training involves two critical challenges: **(i)** Joint training on data from different skills, environments, objects, and even embodiments will add learning difficulty and might lead to optimization conflicts. It is unclear how to fuse the complementary information of different

data within a single unified model. **(ii)** How to ensure both high performance and computational efficiency during model scaling up process.

To address these challenges, we design a new sparsely-gated Mixture-of-Expert (MoE) architecture (Shazeer et al., 2017) to scale up our trajectory model, coined as **Tra-MoE**. Specifically, we integrate several MoE blocks to replace the original transformer blocks. This design enables efficient joint training of most parameters on the large-scale broad out-of-domain data, capturing the complementary patterns for mutual cooperation and the shared knowledge for better generalization. Meanwhile, for different data and different tokens within the same data, our Tra-MoE naturally forms a specialization when activating different experts. To maintain high computational efficiency while scaling up model capacity, we implement a **top-1** gating strategy for token-choice, ensuring constant FLOPs per token. In sparse MoE, some auxiliary losses are commonly used to enhance performance, including the router z-loss (Zoph et al., 2022) for improving training stability and the load-balancing loss (Lepikhin et al., 2020) for balancing expert activations.

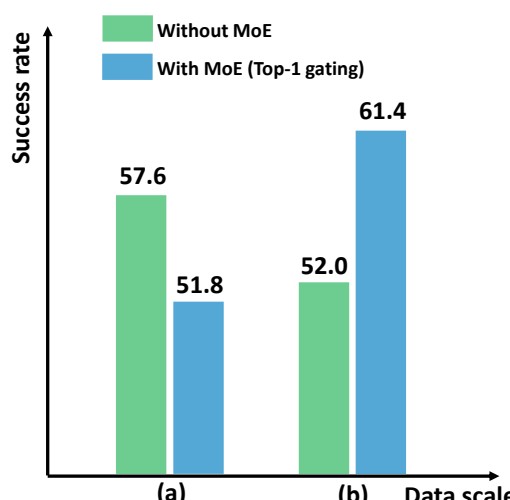

Figure 1: **(a)** Training with small-scale, in-domain data. **(b)** Training with small-scale, in-domain data and large-scale, out-of-domain data.

We conduct a comprehensive empirical study on scaling trajectory models and find that the touter z-loss improves performance while the load-balancing loss tends to cause performance degradation. Furthermore, we observe that when scaling pre-training data from small-scale in-domain to large-scale in-domain and out-of-domain data, employing MoE (Tra-MoE) can lead to obvious performance improvement, whereas the dense counterpart (Tra-baseline) suffers from performance drop, as illustrated in Fig. 1. Notably, even when expanding the dense Tra-baseline's parameter to match that of Tra-MoE, performance improvements remain elusive. These findings collectively demonstrate the effectiveness of our sparse MoE architecture, attributing its capability of balancing parameter cooperation and specialization when scaling up training with out-of-domain data.

Furthermore, given a pre-trained trajectory model, how to effectively condition the policy module using the 2D trajectories still remains an unresolved challenge. We further propose an adaptive policy conditioning technique for trajectory-guided policy. This enables better spatial alignment between 2D trajectories and images, while achieving more flexible trajectory representation and guidance for the policy. Finally, we also conduct real-world experiments to further strengthen our conclusions. In summary, our main contributions can be summarized as follows:

- We propose a sparsely-gated MoE architecture to scale up trajectory models, called as Tra-MoE, with large-scale, out-of-domain, and action-free video data, coupled with a comprehensive empirical study on its training techniques.
- Both simulation and real-world environments demonstrate that our Tra-MoE (with Top-1 gating) can more effectively benefit from out-of-domain data compared to the dense baseline.
- We demonstrate that Tra-MoE still significantly outperform expanded dense counterpart with equivalent parameters, highlighting the effectiveness of sparse MoE architecture in leveraging out-of-domain data through better parameter cooperation and specialization.
- We further propose an adaptive policy conditioning technique for trajectory-guided policy, which ensures adaptive 2D trajectory representation and explicit spatial alignment with the image observations, thereby achieving superior performance.

## 2 RELATED WORK

**Scaling in Robot Learning.** Scaling laws (Kaplan et al., 2020) have fueled substantial advancements in various machine learning fields, including natural language processing and computer vision.

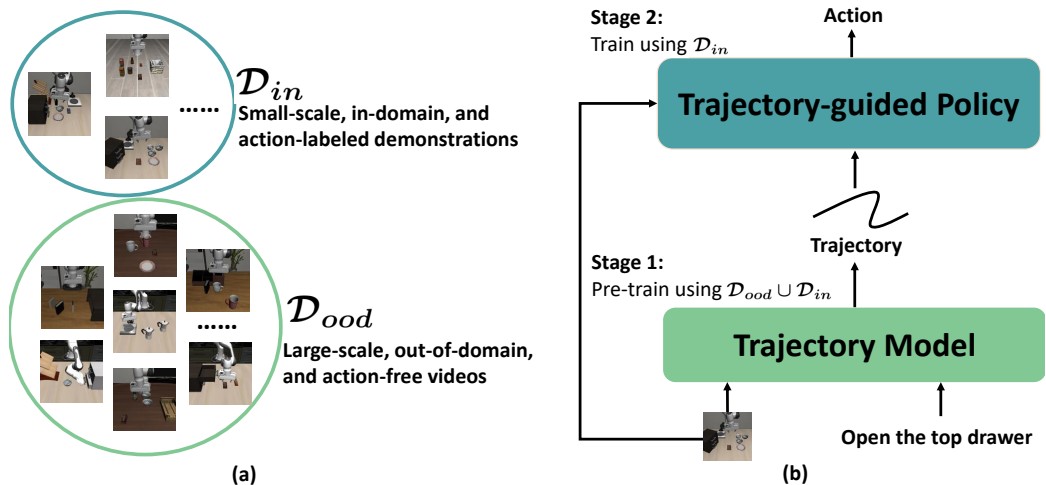

Figure 2: **Scaling up trajectory prediction model**. **(a)** The visualization of Dataset $\mathcal{D}_{ood}$ and $\mathcal{D}_{in}$. The former contains additional skills, environments, objects. **(b)** Our pipeline: first co-train trajectory prediction model and then adapt for downstream policy learning.

In robot learning, many recent studies (Reed et al., 2022; Jiang et al., 2022; Brohan et al., 2022; Bousmalis et al., 2023) investigate how compute, model size, and training data quantity affect the model performance on a range of robotic manipulation tasks. Based on this, recent works have focused on collecting larger-scale real-world robotic datasets (Fang et al., 2023; Walke et al., 2023; Khazatsky et al., 2024), and have successfully trained several robotics foundational models (Brohan et al., 2023; Li et al.; Padalkar et al., 2023; Team et al., 2024; Kim et al., 2024; Doshi et al.) that demonstrate superior transfer capabilities. These works directly combine heterogeneous robotic data with different action and observation spaces for joint training, potentially leading to sub-optimal solutions. In contrast, we leverages sparsely-gated MoE to incorporate larger-scale, out-of-domain, and action-less video data. Furthermore, our experiments demonstrate the importance of MoE for scaling out-of-domain data due to its superior capacity for parameter cooperation and specialization.

**Mixture-of-Experts.** The Mixture-of-Experts (MoE) framework (Cai et al., 2024) is built upon a simple yet effective concept: dividing a model into specialized components, or 'experts', each focusing on distinct tasks or data aspects. This paradigm allows for selective activation of relevant experts based on the input, thereby maintaining computational efficiency while leveraging a vast reservoir of specialized knowledge through increased model capacity. Recently, in computer vision, natural language processing, and multi-modal, there has been a surge of foundation models trained based on the MoE architecture, including Swin-Moe (Hwang et al., 2023), MoE-LLaVA (Lin et al., 2024), DeepSeek-V2 (Liu et al., 2024a), Mixtral-8x22B (Jiang et al., 2024), and so on. MoE typically encompass dense (Puigcerver et al., 2023) and sparse (Chen et al., 2023) variants. Dense MoE activates all experts in each iteration, while sparse MoE activates only a subset, generally resulting in lower computational overhead. However, sparse MoE can suffer from expert load imbalance and training instability issues. Techniques such as adding noise, load balancing loss (Lepikhin et al., 2020), and router z-loss (Zoph et al., 2022) are often employed to mitigate these issues and improve performance. In our work, we conduct a comprehensive empirical study of these techniques when training our sparsely-gated MoE-based trajectory model.

**Representations and Techniques for Policy Conditioning.** Policy conditioning equips a multi-task robot policy with the ability to modulate its behavior by incorporating task-specific guidance information. This information, which may take various forms, guides the policy's action generation, enabling a single model to effectively learn and perform diverse tasks. Classic policy conditioning representations often encompass task identifiers (Rahmatizadeh et al., 2018), natural language instructions (Shridhar et al., 2022; Guhur et al., 2023; Lynch et al., 2023; Brohan et al., 2022), goal images (Lynch et al., 2020), and demonstration videos (Jang et al., 2022; Jain et al., 2024). These are typically integrated with visual observations within the policy through concatenation or attention mechanisms. Recent works have explored alternative policy conditioning representations to enhance generalization and sample efficiency. These include leveraging internet-trained models

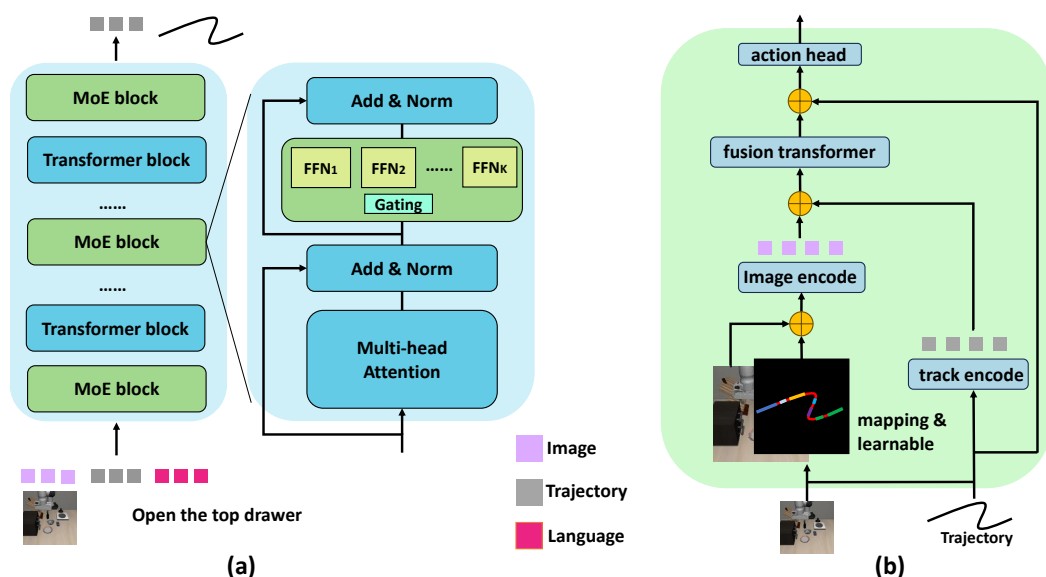

Figure 3: **(a)** The pipeline of our sparsely-gated MoE-based trajectory model (Tra-MoE). **(b)** The pipeline of our trajectory-guided policy using the adaptive policy conditioning technique.

to obtain bounding box (Stone et al., 2023) and object mask (Zhu et al.; Yang et al., 2023), as well as learning plan (Bharadhwaj et al., 2024a), affordance (Bahl et al., 2023), future observations (Du et al., 2024; Du et al.; Yang et al., 2024; Ko et al.), 2D or 3D trajectory (Wang et al.; Wen et al., 2023; Yuan et al., 2024; Bharadhwaj et al., 2024b; Xu et al., 2024) from human video data, and also hand-drawn trajectory (Gu et al.) or sketch (Sundaresan et al., 2024). Different from these approaches that directly utilize raw or manually crafted signals for integration with visual observations, we introduce an adaptive policy conditioning technique that explicitly maps 2D trajectories onto the visual observations in space and encodes them as learnable embeddings. These designs ensure adaptive 2D trajectory representation and explicit spatial alignment with the visual observations.

## 3 METHODOLOGY

In this section, we first describe our problem formulation in Sec. 3.1. Subsequently, we give an overview of our proposed framework, which is composed of a sparsely-gated MoE-based trajectory prediction model and a trajectory-guided policy model using the adaptive policy conditioning technique, detailed in Sec. 3.2. Finally, we describe our training process and loss functions in Sec. 3.3.

### 3.1 PROBLEM FORMULATION

Following the setup of previous works (Wen et al., 2023; Bharadhwaj et al., 2024b; Xu et al., 2024), our approach encompasses a trajectory prediction model to conduct language-conditioned any-point trajectory prediction and a trajectory-guided policy model to predict executable robot actions. In particular, as depicted in Fig. 2, we consider two types of datasets for training, which are large-scale, out-of-domain, and action-free videos $\mathcal{D}_{ood}$, as well as small-scale, in-domain, and action-labeled demonstrations $\mathcal{D}_{in}$. For trajectory prediction model, we employ $\mathcal{D}_{ood}$ and $\mathcal{D}_{in}$ for joint pre-training, whereas for trajectory-guided policy model, we only use $\mathcal{D}_{in}$ for training.

### 3.2 ARCHITECTURE

**ATM Revisited.** We develop our approach based on ATM (Wen et al., 2023), an advanced and promising framework of robotic learning from videos. The overall pipeline of ATM is illustrated on Fig. 2(b). It first learns a trajectory model from video data to perform language-conditioned any-point trajectory prediction. The predicted trajectories provide detailed control guidance, which is then used to learn a trajectory-guided policy model using action-labeled demonstration data. Specifically,

as for a trajectory model, its objective is to predict the future of any-point trajectories in a frame. Formally, given an image observation $o_t$ at timestep $t$, any set of 2D query points simpled on the image frame $\mathbf{p}_t = \{p_{t,k}\}_{k=1}^K$, and a language instruction $\ell$, the trajectory model learns a mapping $\mathbf{p}_{t:t+H} = \tau_\theta(o_t, \mathbf{p}_t, \ell)$ that predicts the coordinates of these query points in the camera frame for the next $H$ time steps, where $\mathbf{p}_{t:t+H} \in \mathbb{R}^{H \times K \times 2}$. For the trajectory-guided policy, it leverages the current observation $o_t$ and the predicted trajectory $\mathbf{p}_{t:t+H}$ to predict the subsequent robot actions.

**Sparsely-gated MoE-based trajectory model (Tra-MoE).** ATM proposes a track transformer to implement the trajectory model. In track transformer, language tokens, image tokens, and track tokens to be decoded involve global interactions within the transformer block. However, directly processing diverse domains and multiple modalities data within a unified transformer may yield suboptimal results. Hence, we propose our Tra-MoE framework. To develop our sparsely-gated MoE-based trajectory model, as depicted in Fig. 3(a), we replace certain layers of the transformer with MoE blocks. In each MoE block, it incorporates multiple feedforward networks (FFNs), each designated as an expert, and utilizes a gating function to activate a selected subset (Top-$K$) of these experts. In this paper, we set $K$ to a fixed value of 1 to ensure that the FLOPs per token remain constant relative to the baseline. Formally, the gating network $\mathcal{G}$, parameterized by $\mathbf{\Theta}$ and typically consisting of a linear-softmax network, yields the output $\mathcal{G}(\mathbf{x_s}; \mathbf{\Theta})$, where $\mathbf{x_s}$ represents our token sequences input. We formulate the sparsely-gated token-choice mechanism as follows:

$$\mathcal{G}(\mathbf{x_s}; \mathbf{\Theta})_i = \text{softmax}(\text{Top-K}(g(\mathbf{x_s}; \mathbf{\Theta}), k))_i, \tag{1}$$

$$\text{Top-K}(g(\mathbf{x_s}; \mathbf{\Theta}), k)_i = \begin{cases} g(\mathbf{x}; \mathbf{\Theta})_i, & \text{condition,} \\ -\infty, & \text{otherwise.} \end{cases} \tag{2}$$

$$\text{condition} : \text{if } g(\mathbf{x}; \mathbf{\Theta})_i \text{ is in the top-}k \text{ elements of } g(\mathbf{x}; \mathbf{\Theta}). \tag{3}$$

More specifically, Top-K$(\cdot, k)$ function retains only the top-$k$ values in a vector, setting the rest to $-\infty$. Consequently, the output of the sparsely-gated MoE layer can be formulated as:

$$\mathcal{F}_{\text{sparse}}^{\text{MoE}}(\mathbf{x_s}; \mathbf{\Theta}, \{\mathbf{W}_i\}_{i=1}^K) = \sum_{i=1}^K \mathcal{G}(\mathbf{x_s}; \mathbf{\Theta})_i f_i(\mathbf{x_s}; \mathbf{W}_i), \tag{4}$$

where $f_i$ represents the i-th expert, usually a linear-GELU-linear FFN layer, is parameterized by $\mathbf{W}_i$.

**Trajectory-guided policy using the adaptive policy conditioning technique.** After completing the trajectory model learning, we develop our trajectory-guided policy model through the adaptive policy conditioning technique in accordance with the initial ATM framework, as shown in Fig. 3(b). In our policy model, independently encoded track tokens and image tokens undergo a early fusion within the fusion transformer. Subsequently, the fused features are further integrated with the raw trajectory data through a late fusion mechanism along the channel dimension, ensuring the preservation of original trajectory information for more accurate action prediction.

Fundamentally, the policy model bridges the gap between the predicted 2D any-point trajectories and the executable 3D robot actions, functioning as an inverse dynamic model and eliminating the necessity of other task specification. However, how to effectively condition the policy using the predicted 2D trajectories remains an unresolved challenge. Based on our analysis, we present two key insights: **(i)** Explicit spatial alignment and fusion of the 2D trajectories with image observations turns out to be highly beneficial. It simplifies the mapping of 2D trajectories to image space and highlights motion-relevant regions, enabling the policy model to focus on important areas and reducing the learning difficulty. **(ii)** Trajectory positions exhibit different characteristics: starting points typically emphasize local motion, while endpoints focus more on global trend. Therefore, an adaptive trajectory representation and conditioning technique is expected to flexibly guide the policy by effectively capturing these variations.

Based on the aforementioned analysis, we propose an adaptive policy conditioning technique to incorporate trajectory information into 2D images. Specifically, we construct an additional mask modality, and populate it with 2D trajectories based on their specific spatial locations, while setting the value of each point in the trajectory as a learnable embedding. Before encoding the image, we concatenate this mask modality with the image observations on the channel dimension, forming a tensor of dimensions $H \times W \times 4$, as shown in Fig. 3(b). To sum up, our adaptive policy conditioning

technique yield an adaptive 2D trajectory representation, which could be explicitly aligned with the image observations to guide the policy prediction more flexibly.

### 3.3 TRAINING

We argue that trajectory models could be viewed as general foundation model, which can be shared by data across multiple domains. Therefore, we first propose to pre-train a generalizable trajectory model on $\mathcal{D}_{ood} \cup \mathcal{D}_{in}$. The $\mathcal{D}_{ood}$ typically contains additional skills, environments, objects, and even embodiments. Then we train the trajectory-guided policy model on $\mathcal{D}_{in}$ based on the pre-trained trajectory model. In trajectory-guided policy training phase, we freeze the trajectory model.

**Pre-training sparsely-gated MoE-based trajectory model.** Following ATM, we train the trajectory model using the ground truth tracks generated by CoTracker (Karaev et al., 2023). We use MSE to optimize the trajectory prediction loss $\mathcal{L}_{\text{trajectory}}$. Additionally, we also consider adding an auxiliary load-balancing loss (Lepikhin et al., 2020) to balance the activation frequency among experts. Given a batch of queries $\mathcal{B}$, it contains B samples and each sample contains S tokens. We define $\mathcal{Q}_i$ as the proportion of tokens distributed to expert $i$, and $\mathcal{P}_i$ as the proportion of the gating probability assigned to expert $i$. The $\mathcal{L}_{\text{load-balancing}}$ can be formulated as follows:

$$\mathcal{Q}_i = \frac{1}{S \cdot B} \sum_{x_s^m \in \mathcal{B}} \mathbb{1}\{\arg\max \mathcal{G}(\mathbf{x_s^m}; \mathbf{\Theta}) = i\}, \tag{5}$$

$$\mathcal{P}_i = \frac{1}{S \cdot B} \sum_{x_s^m \in \mathcal{B}} \mathcal{G}(\mathbf{x_s^m}; \mathbf{\Theta})_i, \tag{6}$$

$$\mathcal{L}_{\text{load-balancing}} = N \sum_{i=1}^{N} \mathcal{Q}_i \mathcal{P}_i. \tag{7}$$

Additionally, ST-MoE (Zoph et al., 2022) proposed using the router z-loss to penalize large logits entering the gating network, thereby improving training stability. We similarly applies it to the training of our trajectory model, as shown below:

$$\mathcal{L}_z(x) = \frac{1}{S} \sum_{k=1}^{S} \left( \log \sum_{i=1}^{N} e^{\mathbf{g}_i^{(k)}} \right)^2, \tag{8}$$

where $S$ is the total number of tokens, $N$ is the total number of experts, and $\mathbf{g} \in \mathcal{R}^{S \times N}$ are the logits going into the router. In summary, the trajectory model's training loss is as follow:

$$\mathcal{L}_{\text{total}} = \lambda_{\text{trajectory}} \cdot \mathcal{L}_{\text{trajectory}} + \lambda_{\text{load-balancing}} \cdot \mathcal{L}_{\text{load-balancing}} + \lambda_z \cdot \mathcal{L}_z. \tag{9}$$

**Training trajectory-guided policy through adaptive policy conditioning technique.** For trajectory-guided policy training, we employ the paradigm of behavior cloning by learning from demonstrations (Lfd). Specifically, we use the MSE between predicted actions and ground truths as the loss function.

## 4 EXPERIMENTS

In this section, we perform experiments using the LIBERO benchmark (Liu et al., 2024b) and a real low-cost dual-arm robot (Koch, 2024). Our experiments aim to study the following questions:

**Q1:** Does the use of techniques like router z-loss, load-balancing loss, and adding noise to gating logits enhance the training of trajectory models (Tra-MoE) that utilize sparsely-gated MoE?

**Q2:** Does sparsely-gated Tra-MoE, compared to the dense Tra-baseline, benefit more from large-scale out-of-domain data and demonstrate superior scaling up capabilities?

**Q3:** When we scale up the dense Tra-baseline to match the parameter count of Tra-MoE, does the sparsely-gated Tra-MoE still maintain its advantage?

Table 1: The ablation experiments on the challenging LIBERO benchmark.

(a) The value of $\lambda_z$.

|  | Spatial | Goal | Object | Long | Avg. |
|---|---|---|---|---|---|
| 0 | 62.0 | 73.0 | 71.0 | 21.5 | 56.9 |
| $5 \cdot 10^{-4}$ | 69.5 | 73.5 | 68.0 | 22.5 | 58.4 |
| $1 \cdot 10^{-4}$ | 62.5 | 81.0 | 73.5 | 28.5 | **61.4** |
| $1 \cdot 10^{-5}$ | 60.0 | 65.5 | 68.5 | 27.0 | 55.3 |
| $1 \cdot 10^{-6}$ | 60.0 | 79.0 | 53.0 | 31.5 | 55.9 |

(b) The value of $\lambda_{load-balancing}$.

|  | Spatial | Goal | Object | Long | Avg. |
|---|---|---|---|---|---|
| 0 | 62.0 | 73.0 | 71.0 | 22.5 | 56.9 |
| $1 \cdot 10^{-3}$ | 58.5 | 74.0 | 60.0 | 18.0 | 52.6 |
| $1 \cdot 10^{-5}$ | 61.0 | 69.5 | 56.0 | 26.0 | 53.1 |
| $1 \cdot 10^{-7}$ | 66.5 | 67.0 | 73.5 | 22.0 | **57.3** |

(c) Adding a noise term to the gating logits.

|  | Spatial | Goal | Object | Long | Avg. |
|---|---|---|---|---|---|
| w/o. noise | 62.0 | 73.0 | 71.0 | 22.5 | **56.9** |
| w. noise | 71.5 | 67.0 | 57.5 | 27.0 | 55.8 |

(d) Scaling Tra-baseline from width and depth.

|  | Spatial | Goal | Object | Long | Avg. |
|---|---|---|---|---|---|
| MoE | 62.5 | 81.0 | 73.5 | 28.5 | **61.4** |
| Width | 66.5 | 50.5 | 61.5 | 15.0 | 48.4 |
| Depth | 61.5 | 75.5 | 46.5 | 26.5 | 52.5 |

(e) The number of experts.

| Num. | Spatial | Goal | Object | Long | Avg. |
|---|---|---|---|---|---|
| 1 | 49.5 | 67.0 | 56.5 | 35.0 | 52.0 |
| 2 | 59.0 | 72.5 | 65.5 | 27.0 | 56.0 |
| 3 | 64.0 | 68.0 | 71.0 | 17.5 | 55.1 |
| 4 | 62.0 | 73.0 | 71.0 | 21.5 | **56.9** |

(f) Scaling with MoE and OOD data.

| MoE | OOD | Spatial | Goal | Object | Long | Avg. |
|---|---|---|---|---|---|---|
|  |  | 67.5 | 68.5 | 68.0 | 26.5 | 57.6 |
| ✓ |  | 63.0 | 64.0 | 60.5 | 19.5 | 51.8 |
|  | ✓ | 49.5 | 67.0 | 56.5 | 35.0 | 52.0 |
| ✓ | ✓ | 62.5 | 81.0 | 73.5 | 28.5 | **61.4** |

**Q4:** Is our proposed adaptive policy conditioning technique for trajectory-guided policies more effective than other baseline methods?

## 4.1 SIMULATION EXPERIMENTS

**Experiment setup.** We conduct simulation experiments using the LIBERO (Liu et al., 2024b) benchmark, which is divided into five categories: LIBERO-Spatial, LIBERO-Object, LIBERO-Goal, LIBERO-Long, and LIBERO-90. LIBERO-90 includes 90 tasks, while each of the other categories contains 10 tasks. Accompanied by skilled human demonstrations, these tasks are designed as follows: LIBERO-Spatial focuses on identical objects across varying layouts; LIBERO-Object emphasizes the same layouts with a change in objects; LIBERO-Goal retains the same object types and layouts but introduces dissimilar goals; LIBERO-Long encompasses a broad range of object types and layouts alongside extended task objectives; Lastly, LIBERO-90 presents a highly varied selection of object categories, layouts, environments, and task goals. Specifically, we select 90 tasks from LIBERO-90 to construct dataset $\mathcal{D}_{ood}$, utilizing 20 action-free videos per task; and we also choose 40 tasks from LIBERO-Spatial, LIBERO-Object, LIBERO-Goal, and LIBERO-Long to build dataset $\mathcal{D}_{in}$, using 10 action-labeled demonstrations per task. In summary, for training Tra-MoE, we use a total of 2200 action-free videos, with an out-of-domain to in-domain data ratio of $9 : 2$. For the trajectory-guided policy, we train a separate multi-task policy for each category: LIBERO-Spatial, LIBERO-Object, LIBERO-Goal, and LIBERO-Long. Each policy undergoes behavioral cloning training with 100 action-labeled demonstrations from 10 tasks.

**Implement details.** Building on the implementation of ATM (Wen et al., 2023), we develop our sparsely-gated Tra-MoE and adaptive policy conditioning technique. Specifically, we replace the first, last, and middle layers of the track transformer with MoE blocks. Unless otherwise specified, we set the number of experts to 4 by default and use a **top-1** gating strategy for token choice to ensure the FLOPs per token remain constant. The default Tra-baseline and Tra-MoE have 13.0 M and 23.6 M parameters, respectively. Each expert contains approximately 3.5 M parameters. Please see the appendix for more detailed training and evaluation information.

**Experiment results and analysis.** We present empirical simulation results to address the aforementioned research questions and make the following findings:

**Result 1:** In Tab. 1a, we examine the effect of the router z-loss weight $\lambda_z$. The results indicate that a higher $\lambda_z$ can effectively enhance performance, with an improvement of 4.5 (61.4 vs. 56.9) at $\lambda_z = 1 \cdot 10^{-4}$. In Tab. 1b, we study the impact of the load-balancing loss weight $\lambda_{load-balancing}$. The results show a significant performance decline with larger values of $\lambda_{load-balancing}$, with only a

Table 2: The ablation experiments of the adaptive policy conditioning technique.

|  | Spatial | Goal | Object | Long | Average |
|---|---|---|---|---|---|
| ATM | 62.5 | 81.0 | 73.5 | 28.5 | 61.4 |
| + hand-drawn mask | 69.0 | 58.0 | 85.5 | 33.5 | 61.5 |
| + adaptive mask | 72.5 | 74.0 | 86.0 | 34.5 | **66.8** |

Table 3: The real-world ablation experiments.

| moe | ood data | Reaching-cube | Folding-cloth | Average |
|---|---|---|---|---|
|  |  | 50.0 | 15.0 | 32.5 |
|  | ✓ | 50.0 | 40.0 | 45.0 |
| ✓ | ✓ | 65.0 | 45.0 | **55.0** |

slight improvement at $\lambda_{load-balancing} = 1 \cdot 10^{-7}$. In Tab. 1c, we verify the effectiveness of adding noise to the gating logits, which results in a performance decrease of by 1.1 (from 56.9 to 55.8).

***Finding 1:*** Based on the above results, we find that using router z-loss to penalize large logits entering the gating network can effectively improve training stability and lead to performance gains. Meanwhile, the load-balancing loss used to balance the activation frequency among experts and the addition of noise to enhance expert allocation exploration cannot bring effective performance improvements. An intuitive explanation is that the load-balancing loss might force experts to set shared parameters on data with large domain gap, which could lead to mutual counteraction of learning gradients. This undermines the advantage of parameter specialization in sparsely-gated MoE. Similarly, adding noise is also detrimental to learning parameter specialization among experts.

***Result 2:*** We comprehensively investigate the impact of introducing sparsely-gated MoE and large-scale out-of-domain data on scaling up trajectory models in Tab. 1f. Firstly, when training only with in-domain data, Tra-baseline achieves an average success rate of 57.6, while Tra-MoE only achieves an average success rate of 51.8. Secondly, when adding out-of-domain data to jointly train Tra-baseline with in-domain data, the average success rate drops to 52.0 (from 57.6 to 52.0), with particularly significant performance declines in LIBERO-Spatial and LIBERO-Object. However, when adding out-of-domain data to jointly train Tra-MoE with in-domain data, the average success rate increases to 61.4 (from 51.8 to 61.4). Finally, in Tab. 1e, we also explore the impact of the number of experts in sparsely-gated MoE on performance. The results show that using just two experts can lead to an increase of 4.0 (52.0 vs. 56.0) in the average success rate. With additional experts, performance generally trends upward, albeit with some fluctuations.

***Finding 2:*** Based on the above experimental results, we have the following findings and analysis: **(i)** When the data scale is small, naively expanding the model capacity through sparsely-gated MoE does not lead to performance improvements. A reasonable explanation is that the larger model may lead to a overfitting problem when training on a small-scale dataset. Moreover, when the data scale is small, training sparsely-gated MoE usually results in some experts not being sufficiently trained, thereby affecting performance. **(ii)** Naively introducing out-of-domain data for joint training often leads to performance degradation in the target domain. However, simultaneously incorporating out-of-domain data and scaling up models with sparsely-gated MoE improves performance. This phenomenon suggests that although out-of-domain data generally enhances model generalization, it could hinder task-specific performance in the target domain due to insufficient in-domain data learning. In contrast, expanding the model with out-of-domain data using sparsely-gated MoE achieves a more optimal balance between parameter cooperation and specialization across diverse data and even different tokens within the same data input. This design allows these large-scale data to jointly train most of parameters, ensuring the capture of general patterns for mutual cooperation. **(iii)** Even with a few experts, Tra-MoE exhibits significant scaling up capabilities. Additionally, further increasing the number of experts generally leads to performance improvement. However, this improvement is not guaranteed due to factors like data-expert complexity matching and training stability.

***Result 3:*** In Tab. 1d, we further investigate the performance comparison between the dense Tra-baseline and the sparse Tra-MoE with the same number of parameters. Specifically, we expand Tra-baseline to match the model capacity of Tra-MoE by increasing both its model width and depth

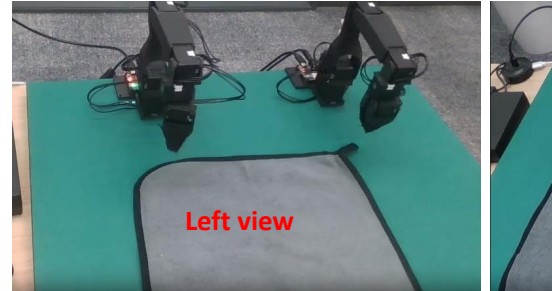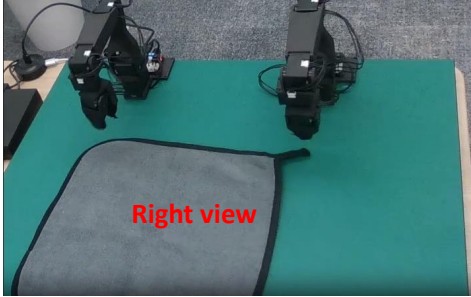

Figure 4: The setup of our real-world experiments. We use two camera views (left and right) to ensure better environment perception.

separately. These expanded versions achieve average success rates of 48.4 and 52.5 respectively, which are significantly lower than Tra-MoE's 61.4.

***Finding 3:*** Based on these results, we are surprised to find that simply expanding the dense Tra-baseline in depth and width can not effectively improve performance. This highlights the importance of the sparsely-gated MoE architecture when training with large-scale out-of-domain data, as it ensures dynamic activation of different experts based on different data input and different tokens within the same data input, significantly reducing optimization conflict problems.

***Result 4:*** In Tab. 2, we investigate the effectiveness of the adaptive policy conditioning technique. Specifically, building upon ATM, we introduce an additional hand-drawn mask modality before image encoding. In this mask, the first half of the trajectory is set to 128, while the second half is set to 255. We call it 'ATM + hand-drawn mask'. Next, we replace the hand-drawn mask modality with an adaptive mask modality, where each point on the trajectory is instantiated as a learnable embedding. We call it 'ATM + adaptive mask'. The results show that 'ATM + hand-drawn mask' achieves only a 0.1 improvement compared to the baseline. However, except for a significant performance decrease in LIBERO-Goal, it achieves performance improvements of 6.5, 12.0, and 5.0 in LIBERO-Spatial, LIBERO-Object, and LIBERO-Long respectively. In addition, the 'ATM + adaptive mask' achieves a performance improvement of 5.4 (66.8 vs. 61.4) compared to the baseline. It further improves performance across all suites compared to 'ATM + hand-drawn mask', especially achieving a 16.0 performance gains in LIBERO-Goal (74.0 vs. 58.0).

***Finding 4:*** Based on these results, we find that explicitly mapping 2D trajectories to visual observations significantly improves the performance of LIBERO-Spatial, LIBERO-Object, and LIBERO-Long, but causes a notable decrease in LIBERO-Goal. By instantiating the points in the 2D trajectory as learnable embeddings, we not only further enhances the performance of LIBERO-Spatial, LIBERO-Object, and LIBERO-Long, but also significantly improves LIBERO-Goal. We analyze that the decline in LIBERO-Goal is mainly due to overlapping front and back parts of some trajectories in the 2D images, leading to mapping errors. However, our adaptive trajectory representation and conditioning technique effectively mitigates this issue. In summary, these experiments collectively demonstrate the effectiveness of explicitly aligning 2D trajectories with images in space as well as applying the adaptive conditioning technique.

### 4.2 REAL-WORLD EXPERIMENTS

**Experiment setup.** As depicted in Fig. 4, we utilize a open-sourced low-cost dual-arm robot Koch (2024) to conduct real-world experiments. Additionally, based on two RealSense cameras, our real-world scenario includes two camera viewpoints: left and right. We evaluate two real-world tasks: a single-arm reaching-cube task and a dual-arm folding-cloth task. The former requires the end-effector of the right arm to successfully contact a red cube on the table, while the latter requires the left and right arms to cooperatively fold a towel on the table. Please see the appendix for more detailed evaluation details and real-world setup.

**Implement details.** Following our simulation experiments, we utilize two camera views and uniformly resize their resolution to 128×128 for trajectory model and policy training. We train a two-expert Tra-MoE (16.3M parameters) on video data from two tasks, with a separate policy trained

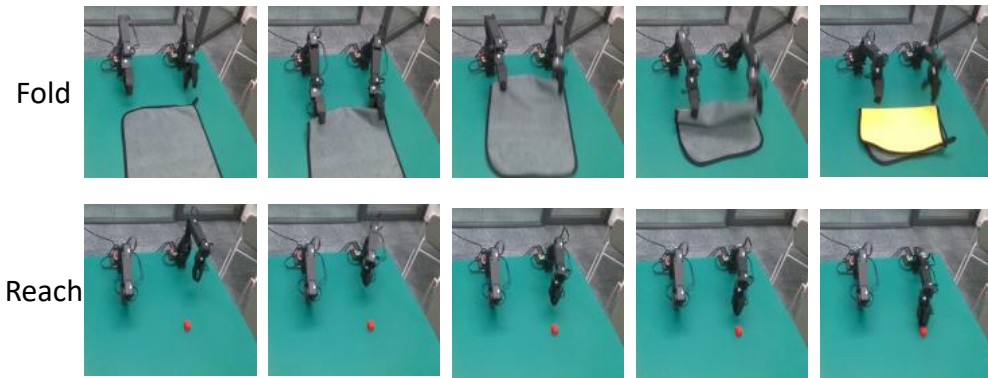

Figure 5: The evaluation demonstrations of our real-world tasks. We evaluate a single-arm reaching-cube and a dual-arm folding-cloth task.

for each task. Therefore, our real-world Tra-MoE integrates data across different embodiments (single-arm and dual-arm). The evaluation demonstrations of our real-world tasks is shown in Fig. 5.

**Experiment results and analysis.** To further strengthen our conclusion that sparsely-gated MoE can better scale up models and out-of-domain data, we conduct real-world experiments. Specifically, for Reaching-cube video data, Folding-cloth is considered out-of-domain data as it involves different embodiments and skills, and similarly for the reverse case. As shown in Tab. 3, due to the limited amount of real-world video data used, even simply adding out-of-domain data to train the trajectory model improves in-domain task performance, especially for the Folding-cloth task. Moreover, further applying MoE (Tra-MoE) leads to additional performance gains. This demonstrates the effectiveness of our sparsely-gated MoE architecture in scaling up out-of-domain data and model capacity.

## 5 CONCLUSION

In this work, we have introduced **Tra-MoE**, a novel framework that utilizes a sparsely-gated MoE architecture with **Top-1** gating strategy to scale up trajectory models. Tra-MoE achieves superior parameter cooperation and specialization, enabling more effective utilization of large-scale out-of-domain data while maintaining constant FLOPs per token. We train our Tra-MoE on all 130 tasks in the LIBERO benchmark and conduct a comprehensive empirical study, demonstrating that Tra-MoE consistently exhibits superior performance compared to the dense baseline model, even when the latter is scaled to match Tra-MoE's parameter count. Additionally, we propose an adaptive policy conditioning technique for trajectory-guided policies, ensuring adaptive 2D trajectory representation and explicit spatial alignment with image observations, thereby achieving superior performance. Extensive experiments in both simulated and real-world environments validate our findings and demonstrate the effectiveness of our proposed methods.

## 6 LIMITATIONS AND FUTURE WORK

In our work, we demonstrate the effectiveness of using sparsely-gated Mixture-of-Experts (MoE) for comprehensively scaling model capacity and leveraging large-scale out-of-domain data, while conducting a thorough experimental study of its training techniques. To the best of our knowledge, this is the first attempt of using sparse MoE architecture for scaling up in robot learning. We hope it can serve as a strong baseline and facilitate further research in this direction. While we are encouraged by the strong results across a wide range of simulated and real-world experiments, a number of limitations and future works still remain. On one hand, we plan to further scale up our Tra-MoE by incorporating more diverse simulation environments, different embodiments, and even real-world human videos. On the other hand, with the increasing availability of large-scale robotic dataset (Padalkar et al., 2023), our sparse MoE architecture could be directly applied to train end-to-end policies, potentially leading a powerful robotic foundation model, akin to Octo (Team et al., 2024) and OpenVLA (Kim et al., 2024). Finally, as for our adaptive policy conditioning technique, a further avenue of exploration is to extend it to other policy conditioning representations, such as human hand trajectories (Wang et al.; Mendonca et al., 2023).

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

## A  SIMULATION EXPERIMENTS DETAILS

In this section, we further elaborate on the details of our simulation experiments. The training hyperparameters for the trajectory model and the trajectory-guided policy are shown in Tab. 4 and Tab. 5, respectively. For the majority of the hyperparameters, we inherit the settings from ATM (Wen et al., 2023). Additionally, when we extend Tra-baseline in depth, the depth is increased from 8 to 14; when we extend Tra-baseline in width, the dimension is increased from 384 to 512. Finally, following the original LIBERO (Liu et al., 2024b) setup, we perform 20 trials for each task evaluation, ensuring a total of 800 trials for each model evaluation.

Table 4: Hyperparameters of our trajectory model.

| Hyperparameters | In-domain data | Out-of-domain data |
|---|---|---|
| Number of videos | 400 | 2200 |
| Epoch | 1000 | 300 |
| Batch size | 2048 | |
| Optimizer | AdamW | |
| Learning rate | 1e-4 | |
| Weight decay | 1e-4 | |
| LR scheduler | Cosine | |
| LR warm-up | 5 | |
| Clip grad | 10 | |
| Point sampling | Variance filtering | |
| Number of points | 32 | |
| Track length | 16 | |
| Augmentation | ColorJitter, RandomShift | |
| dropout | 0.2 | |
| depth | 8 | |
| dimension | 384 | |

Table 5: Hyperparameters of our trajectory-guided policy.

| Hyperparameters | Policy |
|---|---|
| Number of demonstrations | 100 |
| epoch | 120 |
| batch size | 384 |
| optimizer | AdamW |
| learning rate | 5e-4 |
| weight decay | 1e-4 |
| lr scheduler | Cosine |
| lr warm up | 0 |
| clip grad | 100 |
| point sampling | grid |
| number of points | 32 |
| track length | 16 |
| frame stack | 10 |
| augmentation | ColorJitter,RandomShift |
| dropout | 0.1 |

## B  REAL-WORLD EXPERIMENTS DETAILS

In this section, we further elaborate on the details of our real-world experiments. Specifically, we use two leader arms to perform teleoperation for data collection, as shown in Fig. 6, where 50 demonstrations are collected for each task for trajectory model and policy training. For our real-world evaluations, we conduct 20 triasl for each task, while ensuring, to the extent possible, that the object

poses in the training set differ from those in the test set. For the relevant training hyperparameters, we maintain consistency with the simulation experiments.

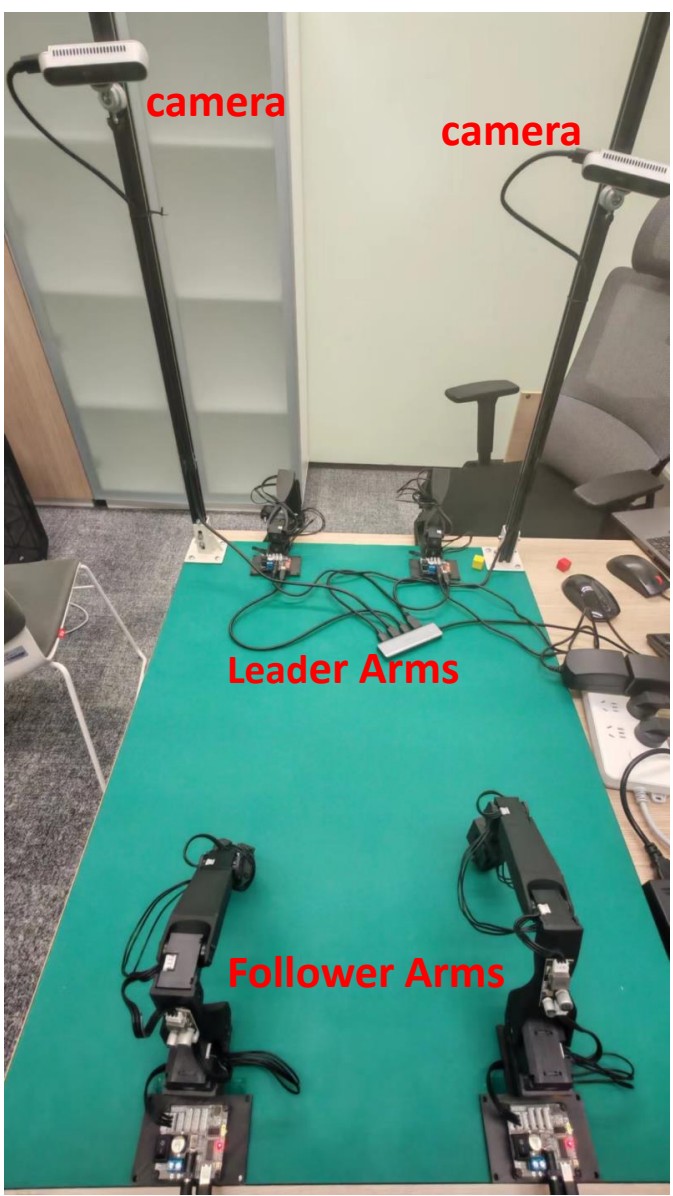

Figure 6: Our real-world hardware platform.

