# OpenReview forum: "Tra-MoE: Scaling Trajectory Prediction Models for Adaptive Policy Conditioning"
_ICLR.cc/2025/Conference — ICLR 2025 Conference Withdrawn Submission_

### Official Review · Reviewer_uBPP · 2024-10-26

**Soundness:** 2
**Presentation:** 3
**Contribution:** 2
**Rating:** 3
**Confidence:** 4

**Summary:**

The paper proposes a sparse-MoE based trajectory prediction model, Tra-MoE,  with the goal of improving trajectory prediction performance compared to prior methods. The authors also propose an adaptive policy conditioning technique to use the 2D trajectory predictions as an input to a trajectory guided policy. The paper evaluates their approach across both simulated and real world environments and provides an ablation analysis on different training techniques used in Tra-MoE.

**Strengths:**

- The authors take inspiration from recent trajectory guided policy learning algorithms that learn a trajectory prediction model on action-free video data to guide the robot policy. Accordingly, they propose a sparse-MoE based trajectory predictor that aims to scale up trajectory models.
- The authors also propose an adaptive policy conditioning technique for trajectory-guided policy, acheiving superior performance than prior methods.
- The paper includes experiments across both simulated and real world environments along with ablation studies to justify design choices.

**Weaknesses:**

- The paper aims to improve trajectory prediction performance compared to prior methods. However, I do not see a comparison between the trajectory prediction performance of Tra-MoE and other methods. It would be great if the authors could include these comparisons, atleast with the trajectory prediction models from recent works like ATM (transformer based) [1] and Track2Act (diffusion-transformer based) [2].
- For the LIBERO experiments, the authors only use 10 or 20 trajectories per task. However, I believe that LIBERO provides 50 demonstrations per task. Is there a reason for using a smaller number of demonstrations? An analysis of performance across varying number of demonstrations for both Tra-MoE and baselines (trajectory predictors from ATM, Track2Act) would help highlight the advantages of using Tra-MoE.
- How do the trajectory-guided policies compare with image-based BC? Is there an advantage of using trajectory guidance? Recent works [3] have shown >80% performance on LIBERO using image-based policies. Hence, it would be interesting to compare with these works to highlight the usefulness of trajectory guidance for policy learning.
- It would be great if the authors could also include real-world policy performance comparisons with the baselines.

References
[1] Wen, Chuan, et al. "Any-point trajectory modeling for policy learning." arXiv preprint arXiv:2401.00025 (2023).
[2] Bharadhwaj, Homanga, et al. "Track2Act: Predicting Point Tracks from Internet Videos enables Diverse Zero-shot Robot Manipulation." arXiv preprint arXiv:2405.01527 (2024).
[3] Haldar, Siddhant, Zhuoran Peng, and Lerrel Pinto. "BAKU: An Efficient Transformer for Multi-Task Policy Learning." arXiv preprint arXiv:2406.07539 (2024).

**Questions:**

It would be great if the authors could address the concerns mentioned in Weaknesses, especially the comparisons with existing methods. I am willing to increase my score once these concerns have been addressed.

---

### Official Review · Reviewer_cD3p · 2024-10-29

**Soundness:** 2
**Presentation:** 2
**Contribution:** 3
**Rating:** 5
**Confidence:** 4

**Summary:**

This paper introduces Tra-MoE, an approach for scaling trajectory prediction models in robot learning.  The key idea is to leverage large-scale, out-of-domain, action-free video data to improve the robustness and generalization of these models. Tra-MoE employs a sparsely-gated Mixture-of-Experts (MoE) architecture, where a top-1 gating strategy activates only one expert per token, maintaining computational efficiency while allowing the model to specialize on different data aspects. The authors also propose an adaptive policy conditioning technique that learns 2D mask representations for predicted trajectories, aligning them with image observations to better guide policy prediction.

Experiments on the LIBERO benchmark and a real-world dual-arm robot demonstrate that Tra-MoE outperforms a dense baseline model, even when scaled to the same parameter count, showing the effectiveness of the sparse MoE architecture and adaptive conditioning in leveraging out-of-domain data for improved performance. The authors also conducted an empirical study of MoE training techniques for trajectory models, showing the benefits of router z-loss and the drawbacks of load-balancing loss and noise addition in this context.

**Strengths:**

- The application of sparsely-gated MoE architecture for trajectory prediction in robot learning demonstrates a practical approach to model scaling while preserving computational efficiency. Though the idea is quite straightforward, it is definitely worth trying. The paper provides an empirical validation of this approach.

- Through extensive empirical analysis of MoE training techniques, including router z-loss, load-balancing loss, and noise addition, the study offers valuable insights into employing sparse MoE models for robotics applications.

- The validation of findings on a low-cost physical robot platform strengthens the practical applicability of the proposed methods.

- The paper also introduces an innovative adaptive policy conditioning technique using learned 2D mask representations for predicted trajectories. This approach enhances spatial alignment between trajectories and images, resulting in more effective policy guidance.

**Weaknesses:**

### Major

- While this paper claims to leverage broad out-of-domain video data for training trajectory models, the experiments primarily utilize data from similar domains. In their experiments, the in-domain datasets comprise LIBERO-Spatial, LIBERO-Object, LIBERO-Goal, and LIBERO-Long, while the out-of-domain dataset is LIBERO-90. Although the goals and objects differ across these datasets, many fundamental factors remain consistent, including robot morphology, controllers, camera pose, lighting conditions, physical parameters, etc. Consequently, the "out-of-domain" dataset shares substantial similarities with the in-domain data. Thus, the experiments actually demonstrate that the MoE can consume more data from similar domains, rather than truly out-of-domain data.

- The main evaluation metric used in this paper appears to be success rate, though this is not explicitly stated by the authors. This metric is presumably obtained by evaluating the adaptive policy shown in Figure 3(b), which is conditioned on the trajectory model. However, the success rate does not directly measure the trajectory model's performance. While better trajectory models may generally lead to improved policy performance, this relationship requires quantitative validation to establish a clear correlation between trajectory model quality and policy success rates.

- According to Table 1(f), incorporating additional "out-of-domain" data and implementing the MoE architecture yields only a modest 3.8% improvement in success rate. This marginal gain, coupled with the fact that success rate is an indirect measure of trajectory model performance, raises questions about the actual benefits of combining MoE architecture with "out-of-domain" data.

- While this paper presents a good empirical study, its methodological novelty and contributions are limited, essentially just combining ATM with MoE as its primary training recipe.


### Minor
- Figure 1 lacks proper annotation of the y-axis scale, which makes it potentially misleading.
- Figure 3(b) is quite confusing. The arrow layout does not clearly explain the information flow.
- The paper's ablation study on the number of experts is limited to a narrow range (1-4), whereas modern MoE architectures [1] typically employ more experts (e.g., 8). A more extensive exploration of how performance scales with the number of experts would provide a more complete understanding of the trade-offs involved.

[1] Jiang, Albert Q., et al. "Mixtral of experts." arXiv preprint arXiv:2401.04088 (2024).

**Questions:**

- Are the numbers listed in Table 1 success rates? If so, could you please define them clearly?

---

### Official Review · Reviewer_ZFqS · 2024-11-01

**Soundness:** 3
**Presentation:** 2
**Contribution:** 2
**Rating:** 3
**Confidence:** 4

**Summary:**

The authors propose an imitation learning method that additionally conditions on predicted future trajectories of arbitrary points in an image. In the first stage a model is trained to predict point trajectories (with labels provided by CoTracker). In the second stage, the policy is trained with the predicted point trajectories as an additional input. In principle, these point trajectories act as a plan to guide the policy and improve performance. Additionally the point trajectory prediction model can be trained on broader action-free datasets to improve generalization. Prior work, Any-point Trajectory Modeling (ATM), has already introduced this method, but this work offers two modifications. They use Mixture-of-Experts (MoE) blocks in the point trajectory model, and they propose a new method for conditioning the policy on the point trajectories where the trajectory is added as an additional channel of the image observation. The claim is that the MoE blocks help the trajectory model learn from more OOD data and the new conditioning mechanism improves policy performance. Experiments in the simulated LIBERO benchmark and on a real robot are used to validate these claims.

**Strengths:**

The paper tests the method on an established simulated benchmark, LIBERO, and on a real robot and finds that the proposed MoE blocks and conditioning mechanism improve performance. Generating intermediate state/action representations for policy conditioning as a means of improving long-horizon planning and generalization is an active area of research. Thus it's likely the community would find these results interesting.

**Weaknesses:**

My main concerns are with respect to the novelty and the evaluation.

**Novelty**

The method explored in this paper was introduced by Any-point Trajectory Modeling [1] and many other papers [2, 3, 4, 5, 6, 7, 8], and the two proposed architectural modifications seem rather incremental. Adding MoE blocks does not make a lot of sense at the model and data scales studied in the experiments (see discussion on evaluation below). Additionally, the proposed conditioning mechanism seems to be almost exactly the same as what was used in RT-Trajectory [2].

**Evaluation**

The claim is that the MoE blocks help with scaling to more diverse and out-of-domain data. However the OOD data used in the experiments is not OOD relative to what has been explored in similar papers (see those cited below). For LIBERO the OOD data is just LIBERO data from other scenes but with the same robot. For the real robot experiments, the OOD data is data from the same robot but a different task. Other papers in this area have explored using data from humans or robots with different morphologies. Given the claim that the MoE blocks help with scaling to significantly more diverse data, I would have expected the experiments to use OOD data that is at least as OOD as what prior work has tried.

Additionally, sparsely-gated MoE was originally proposed as a method for scaling models past billions of parameters, whereas the models studied in the experiments are less than 30M parameters. While parameter efficiency is nice at any model scale, the main challenge/bottleneck for point trajectory prediction methods is not scaling the model. There is an experiment that tries scaling the dense model in either width or depth, and these larger dense models don't perform as well as the MoE model. However, this is only one data point and I would expect scaling the dense model even further would eventually match the MoE model.

[1] https://arxiv.org/abs/2401.00025

[2] https://arxiv.org/abs/2311.01977

[3] https://arxiv.org/abs/2407.15208

[4] https://arxiv.org/abs/2405.01527

[5] https://arxiv.org/abs/2401.11439

[6] https://arxiv.org/abs/2308.15975

[7] https://arxiv.org/abs/2407.08693

[8] https://arxiv.org/abs/2403.03174

**Questions:**

- Is the proposed trajectory conditioning mechanism different from what was used in RT-Trajectory?
- Would scaling the trajectory model further eventually match the performance of the MoE model?

---

### Official Review · Reviewer_xxPt · 2024-11-07

**Soundness:** 2
**Presentation:** 1
**Contribution:** 2
**Rating:** 3
**Confidence:** 4

**Summary:**

The paper proposes Tra-MoE, a trajectory prediction model with a sparsely-gated Mixture-of-Experts (MoE) architecture that scales up trajectory prediction by effectively leveraging out-of-domain data.  The model also consists an adaptive policy conditioning technique. The authors conduct both simulation and real-world experiments to validate their approach.  Experiments demonstrate that the Tra-MoE exhibits superior performance compared to the dense baseline model

**Strengths:**

1. The sparse MoE approach allows effective utilization of more data with parameter specialization.
2. The proposed method of using learnable trajectory masks aligned with image observations is a effective way of enhancing policy guidance.

**Weaknesses:**

I have several concerns regarding the motivation and clarity of the paper's presentation.

1.  The paper emphasizes scaling as a key factor in trajectory modeling. However, it is well understood that increasing data generally improves neural network performance, which I consider common knowledge rather than a novel contribution. If the goal is to explore scaling, the paper’s focus seems to lean more toward leveraging out-of-domain data rather than simply increasing in-domain data or model size. This raises the question: why not focus on expanding in-domain data, which might yield better performance? Additionally, the model's current size (23.6M parameters) is relatively modest by today's standards. It remains unclear whether the proposed approach could scale effectively to the level of large language models trained on vast datasets.

2. The explanation of the mask modality (used for conditioning 2D trajectories) lacks clarity. Key details are missing about its format, structure. Providing a more thorough description or example would significantly improve understanding.

3. The experiments suggest that adding MoE without additional data leads to a performance drop, raising concerns about the scalability of the top-1 gating strategy. Besides, the result implies that more data may primarily prevent overfitting, rather than allowing the model to scale effectively.

4. The "optimization conflicts" mentioned in the introduction are unclear. It would help to clarify what these conflicts entail and how they impact training.

5. The paper lacks a clear definition of trajectory representation at the outset. Moreover, the choice of MSE as the loss function for trajectory prediction is questionable. Trajectory prediction often involves multi-modalities, where generative models might be more appropriate for capturing diverse outcomes.

**Questions:**

see weakness

---

### Author Response · Authors · 2024-11-15

Thank you to all the reviewers for your valuable feedbacks and suggestions. In the month following the initial submission, we have continuously improved the quality of the manuscript. We will include additional experiments and modify our presentation in future submission.

---

### Note · Authors · 2024-11-15

I have read and agree with the venue's withdrawal policy on behalf of myself and my co-authors.